# Quantitative nylon monomerization by the combination of chemical pretreatment and enzymatic hydrolysis using nylon hydrolases

Yuki Shiraishi[1], Dai-ichiro Kato[1]*, Kaito Miyazaki[1], Maina Yonemura[1], Yoko Furuno[1], Risa Yokoyama[1], Yukiko Yokogawa[1], Sho Nonaka[2], Yoshiro Kaneko[2], Keigo Ebata[3], Yuichiro Himeda[4], Seiji Negoro[5]

1 Department of Science, Graduate School of Science and Engineering, Kagoshima University, Korimoto, Kagoshima, Japan, 2 Department of Engineering, Graduate School of Science and Engineering, Kagoshima University, Korimoto, Kagoshima, Japan, 3 Faculty of Fisheries, Kagoshima University, Shimoarata, Kagoshima, Japan, 4 Global Zero Emission Research Center, National Institute of Advanced Industrial Science and Technology, Tsukuba, Ibaraki, Japan, 5 Department of Applied Chemistry, Graduate School of Engineering, University of Hyogo, Himeji, Hyogo, Japan

* kato@sci.kagoshima-u.ac.jp

**Data Availability Statement:** All relevant data are within the manuscript and its Supporting information files.

## Abstract

Nylons, derived from fossil fuels, are widely used for their toughness and flexibility, but they pose environmental concerns due to their low biodegradability. This study explored an efficient method for the monomerization of polymeric nylons, specifically nylon-6 and nylon-6,6, through a combination of chemical pretreatment and enzymatic hydrolysis using two kinds of nylon hydrolases, NylB and NylC (Nyl series enzymes). To break down the strong intermolecular hydrogen bonding between polymer chains of nylon, two pretreatment methods were investigated: homogeneous dispersion and soluble oligomerization induced by acid treatment. Homogeneous dispersion enhances water solubility, while soluble oligomerization reduces the molecular weight. These pretreatments significantly increased the enzyme sensitivity of the nylons, resulting in nearly complete conversion into monomers by Nyl series. Finally the convincing monomerization toward market products such as used fishing nets was also achieved. This study highlights the potential of this methodology for chemical recycling, offering a promising solution for reducing environmental impacts and achieving a circular economy for nylon products.

## Introduction

Nylon, a type of aliphatic polyamide (PA), is a versatile thermoplastic derived from fossil fuels and conjugated by amide bond linkages. Because of its toughness, flexibility, and heat- and chemical-resistance properties, nylon can be found in a diverse range of daily life applications, including textiles, packaging, and fishing nets, and so on. The annual global production of nylon was reported to be more than 8.9 million tons in 2020, and only two nylons, nylon-6 and nylon-6,6, composed more than 90% of this total production. The production of nylon from petroleum has a negative impact on both energy consumption and greenhouse gas

**Funding:** This work was partly supported by a Moonshot Research and Development Program (JPNP18016) commissioned by the New Energy and Industrial Technology Development Organization and Supporting program for interdisciplinary projects in Kagoshima University (DK), and a grant-in-aid for scientific research [Japan Society for Promotion of Science, No. 19K05171 (SN)].

**Competing interests:** The authors have declared that no competing interests exist.

(GHG) emissions [1]. In addition to the production process, postusage problems are also noted. Like that of many synthetic polymers, the low biodegradability of nylon raises concerns about its potential environmental accumulation. With the growing social demands for sustainable development goals (SDGs), achieving a circular economy through recycling is promising for reducing GHG emissions, diminishing reliance on fossil fuels, and minimizing negative impacts on the environment [2]. Nylon-6 and nylon-6,6 are composed of simple monomer units: nylon-6 consists of 6-aminohexanoate (Ahx), while nylon-6,6 is made from hexamethylenediamine (HMD) and adipic acid (AD). These monomer units can be re-polymerized into polymers either in their original form or after lactamization. As a result, chemical recycling technologies that break down polymeric nylons into these monomer units hold significant potential for advancing circular economic systems [3].

The enzymatic approach has high potential as a superior monomerization method. Research on the enzymatic recycling of PET, a kind of polyester, has made rapid progress toward commercial deployment in recent years [4]. However, hydrolyzing polymeric nylons into their monomer units is not an easy task, and industrial and commercial nylon recycling efforts using chemical processes have only recently begun to gain recognition [5]. The toughness and flexibility of nylon, which are unique properties, are derived from complex interwoven structures of crystalline and amorphous regions [6, 7]. This complex nylon structure is determined by the intermolecular hydrogen bonding pattern between polymer chains. The enzymatic depolymerization is affected by its pattern because the strong intermolecular hydrogen bonding network makes nylon insoluble in water and also reduces the amount of exposed free amide bonds that the enzyme can attack.

At present limited examples of nylon biodegradation are known [8–11]. Among them, we have been developing a hydrolytic approach for accessing various aliphatic nylons, including nylon-6 and nylon-6,6 using two kinds of nylon hydrolases, NylB and NylC, which are named the Nyl series enzymes [12, 13]. NylB is an exo-type mode enzyme that cleaves amide bonds at the N-terminus of Ahx-linear oligomers to sequentially produce Ahx monomers [14, 15]. NylC degrades Ahx-cyclic and -linear oligomers with a degree of polymerization greater than three in an endo-type mode and cleaves the internal amide bonds in oligomers to produce dimeric or higher linear oligomers [16, 17]. In particular, a quadruple NylC mutant (NylC--GYAQ) that showed high thermal stability was designated nylon hydrolase because it was found that its high hydrolytic activity succeeded in converting polymeric nylons into oligomers [16]. However, the degradation efficiency of these enzymes for bulk polymeric nylon was limited to only a few percent at best.

The resistance of nylon toward enzymatic hydrolysis is primarily the result of water insolubility. Based on this expectation, we have considered that a method of chemical pretreatment may be effective for increasing the water solubility of nylon and have examined the effects of the following two chemical processes, homogeneous dispersion and soluble oligomerization. Homogeneous dispersion aims to increase the contact area of the nylon surface with water by adjusting the particle size. Because we have already confirmed that increasing the contact area of nylon by thinning it has a positive effect on the enzymatic reaction, expected by dispersing the nylon polymer finely in water by homogeneous dispersion would greatly increase the contact area with the enzyme, thereby increasing the efficiency of hydrolysis [18]. Acid caused soluble oligomerization is an approach aimed at increasing the water solubility of nylon by lowering its molecular weight and increasing the sensitivity of enzymatic reaction.

S. R. Shukla and coworkers reported an example of soluble oligomerization of nylon-6 using some acids such as formic acid, hydrochloric acid and sulfuric acid [19]. However, complete monomerization had not yet been achieved in their work. Therefore, in order to achieve complete monomerization, we conceived to conduct acid treatment followed by enzymatic

**Table 1. A summary of Shukla's methods and our proposed method.**

| Reference | Applicable Nylon | Method | Enzyme Used | Key Findings |
|---|---|---|---|---|
| Shukla et al. [19] | nylon-6 | Acid caused soluble oligomerization | Non | Concern about oligomers remained |
| This study | nylon-6 & nylon-6,6 | Chemical pretreatment (Homogeneous dispersion and/or Acid caused soluble oligomerization) & enzymatic hydrolysis | NylB-DNY & NylC-GYAQ | Achieve quantitative monomerization and confirm the applicability to market product |

hydrolysis using Nyl series. We also examined the relationship between the average molecular weight and the efficiency of enzymatic hydrolysis by the Nyl series, and predicted the degree of oligomer on which the enzyme could hydrolyze. In addition, we decided to try soluble oligomerization of nylon-6,6, which they had not yet investigated. Table 1 provides a comparison between Shukla's methods and the chemo-enzymatic combination method proposed in this article. Here, we introduce that a quantitative monomerization method for both nylon-6 and nylon-6,6 have succeeded, which was not possible by chemical methods alone, using a combination of chemical pretreatment and Nyl series catalyzed enzymatic hydrolysis.

## Experimental

### 1. General methods

**1.1 Nylon.** The nylon-6 pellets used in this study were kindly gifted from Toyobo Co., Ltd. (Tsuruga, Japan). Nylon-6,6 pellets were purchased from Sigma-Aldrich Co., Ltd. The authentic standards for nylon oligomers were prepared following the method described [12]. The used fishing net of bottom trawl was kindly gifted from Mr. Soumei Okamoto, belonging to Higashikushira Fisheries Cooperative, Shibushi, Kagoshima, Japan.

**1.2 Nyl series enzymes.** In this study, two kinds of nylon hydrolases from *Arthrobacter* sp. KI72, which were called Nyl series enzymes, NylB-DNY (a NylB-NylB' hybrid enzyme with G181D/H266N/D370Y substitutions) and NylC-GYAQ (a NylC enzyme with D122G/H130Y/D36A/E263Q substitutions), were tested for their ability to hydrolyze nylon-6 and nylon-6,6 [14, 17]. For these two enzymes, a His-tagged linker was fused to the N-terminal region. Details are described in Supplemental section.

**1.3 Detection of enzyme activity by thin-layer chromatography (TLC).** The analysis of reaction products released into soluble fractions was carried out using thin-layer chromatography (TLC), as detailed in our prior studies [12]. The supernatant (2 μL) was spotted on a silica gel plate (1.05748; Merck Co. Ltd., Darmstadt, Germany). The samples were developed by a solvent mixture (1-propanol/water/ethyl acetate/ammonia = 24:12:4:1.5), and the degradation products were identified through visualization with a 0.2% ninhydrin solution (in 1-butanol saturated with water).

**1.4 Quantification of amino groups via a colorimetric method.** In a 96-well plate, 6 μL of the sample was added to 75 μL of 0.1 M sodium tetraborate solution (pH 9.5) [20]. Then, 30 μL of TNBS solution (0.05% trinitrobenzene sulfonic acid sodium salt (TNBS) containing 0.065% sodium sulfite) was combined, and the solution was rapidly mixed. After incubating for 60 minutes at 40°C, the absorbance at 420 nm was determined, and the concentration of the released amino groups was estimated via comparison with the absorbance of a known concentration of the 6-aminohexanoic acid (Ahx) standard [12].

**1.5 Particle size measurement.** The average particle sizes and particle size distributions of the homogeneous dispersed nylon samples were determined using a laser diffraction and scattering instrument (LMS-2000e, Seishin Enterprise Co., Ltd., Tokyo, Japan).

**1.6 Gel permeation chromatography (GPC) analysis.**   GPC was carried out at 40˚C on a Toso gel permeation chromatograph system (HLC-8420GPC) equipped with two TSKgel SuperMultiporeHM-Ns. HFIP containing 5 mmol/L TFA-Na was used as the solvent. The flow rate was 0.2 ml/min. The sample concentration and the injection volume of the solution were 1 mg/ml and 10 μl, respectively. EasiVialPM (Agilent, PL2020-0201) was used as a polymer standard.

**1.7 Thermogravimetric analysis (TGA).**   TGA was performed using TGA-50 (SHIMADZU Co., Kyoto, Japan). The samples were heated from room temperature (ca. 25˚C) to 1,000˚C at a heating rate of 10˚C/min under nitrogen flow (100 mL/min).

**1.8 Liquid chromatography-mass spectrometry (LC-MS) analysis.**   LC-MS analysis was performed using an Alliance W2690/5 system (Nihon Waters K.K., Tokyo, Japan) equipped with a PDA Detector W2998 and an ACQUITY QDA Detector operating in positive ion mode. Separation was achieved on an XBridge C18 column (3.5 μm, 4.6 mm × 150 mm, Waters) maintained at 40˚C. The mobile phase consisted of Solvent A (10 mM ammonium acetate) and Solvent B (acetonitrile). A gradient elution program was employed as follows: 0–10 min, 0–15% B; 10–20 min, 15–100% B; 20–30 min, 100% B; 30–40 min, 0% B. The flow rate was set at 0.5 mL/min.

**1.9 1H-NMR measurement.**   The 1H-NMR measurements were conducted using a JNM-ECA600 nuclear magnetic resonance spectrometer (JEOL JNM-ECA600KS, JEOL DATUM LTD., Tokyo, Japan). The sample was ultrafiltered to remove the Nyl series proteins, lyophilized, and subsequently dissolved in deuterium oxide ($D_2O$) to prepare for analysis.

## 2. Homogeneous dispersion of nylons in aqueous solution

A 25 mg/mL nylon solution, which was dissolved in 2,2,2-trifluoroethanol (TFE) at 75˚C, was added dropwise into a vigorously agitated solvent. The ratio of TFE solution-to-solvent was 1:9. Six kinds of solvents, water, ethyl acetate (EtOAc), acetone, methanol (MeOH), ethanol (EtOH), and ethylene glycol (EG), were employed. The suspension was centrifuged at 14,000 revolutions per minute (rpm) for 10 minutes at 20˚C. The resulting pellet was washed with 500 μL of buffer A (20 mM potassium phosphate buffer (pH 7.3) containing 10% glycerol) and centrifuged to remove the supernatant by pipetting. This washing process was repeated three times to eliminate any residual suspended solvent, and was resuspended in buffer A to adjust the concentration to 10 mg/mL for the enzymatic reaction described in the section 5.

## 3. Soluble oligomerization of polymeric nylons by acid hydrolysis

**3.1 Formic acid caused soluble oligomerization.**   To 6.8 mL of 90% formic acid was added 1 g of nylon (150 g/L), and the mixture was stirred in an oil bath at 75˚C, 100˚C or 120˚C. The incubation time was 96 hr, except at 100˚C, for a duration of 168 hr. Approximately 25 mL of distilled water was poured into the reaction mixture, and the mixture was lyophilized to remove formic acid. The dried sample was resuspended in buffer A, the pH was adjusted to 7.3 with 1 M NaOH if necessary, and the solution was diluted to 10 mg/mL for the enzymatic reaction described in the section 5.

**3.2 Formic acid removal from reaction vessel after acid hydrolysis.**   Instead of lyophilization process, another method using a formic acid decarboxylation catalyst, Ir complex containing amino-functionalized ligand ($[Cp^*Ir(4,4'-(NH_2)_2-bpy)(H_2O)]SO_4$), was investigated [21]. This catalyst breaks formic acid down in $CO_2$ and $H_2$. After soluble oligomerization described in the section 3.1, the solution was diluted 2-fold with 6.8 mL of distilled water to adjusted 45% formic acid concentration, and then 1 mg of the formic acid decarboxylation catalyst was added. After the reaction was stirred at 70˚C on overnight, the pH was adjusted to

7.3 by 1 M NaOH, and the solution was diluted to 10 mg/mL for the enzymatic reaction described in the section 5.

**3.3 Other acids caused soluble oligomerization.** To 6.8 or 5 or 2 or 1 mL of 18% hydrochloric acid (2-fold diluted conc. HCl) or 50% sulfuric acid (2-fold diluted conc. $H_2SO_4$) was added 1 g of nylon (150, 200, 500, 1,000 g/L), and the mixture was stirred in an oil bath at 120°C. The heating time was 2 hr. After cooling to room temperature, the pH of reaction mixture was adjusted to 7.3 by 10 M NaOH, and the solution was diluted to 10 mg/mL for the enzymatic reaction described in the section 5.

## 4. Combination of homogeneous dispersion and formic acid caused soluble oligomerization

One gram of nylon, which was homogeneously dispersed in ethanol, was subjected to formic acid caused soluble oligomerization in an oil bath at 100°C for 168 hr. After adding 25 mL of distilled water, the mixture was freeze-dried. The dried sample was resuspended in buffer A for enzymatic hydrolysis. The dried sample was resuspended in buffer A, the pH was adjusted to 7.3 with 1 M NaOH if necessary, and the mixture was diluted to 10 mg/mL for the enzymatic reaction described in the section 5.

## 5. Enzymatic reaction

To 200 μL of the above prepared chemical pretreatment nylon (homogeneous dispersion, soluble oligomerization, or their combination) solution (10 mg/mL) was added 200 μL of NylB-DNY and/or NylC-GYAQ enzyme solution (2.0 mg/mL each in BufferA). The mixture was incubated at 37°C for 24 or 72 hr. During the reaction, sampling (30 μL aliquots taken and heated at 99°C for 10 min for inactivation of enzymatic activity) was performed at regular intervals, and the released amino group concentration was evaluated by TLC analysis and the colorimetric method using TNBS. The monomeric rate is the ratio of the inferred amount of exposed amino groups to the concentration of amino groups in the system. It was calculated by Eq 1. The denominator represents the theoretical maximum amino group concentration when nylon is completely hydrolyzed and all amino groups in nylon are exposed. The numerator is the concentration of the released amino group in the reaction mixture measured by the TNBS method.

$$\text{Monomeric rate}(\%) = \frac{\text{The released amino group concentration}}{\text{The theoretical amino group concentration}} \times 100 \qquad (1)$$

## Results and discussion

### Homogeneous dispersion of nylons in aqueous solution

There are several reports on the nylon-4 or poly(tetramethylene succinate)-co-(tetramethylene adipate) (PBSA) dispersions in aqueous solution [22, 23]. Recently, a technique for achieving fine dispersions of nylon-6 in aqueous solutions using formic acid has been developed, demonstrating enhanced sensitivity to enzymatic hydrolysis [24]. However, formic acid raises concerns about cleaving amide bonds in polymeric nylon, leading to a reduction in molecular weight. We explored the use of TFE, an alternative nylon-dissolving solvent with minimal risk of amide bond cleavage, for creating aqueous dispersions. After some investigation, a very simple and convenient method could be established. After dissolving nylon-6 in 2,2,2-trifluoroethanol (TFE) as a good solvent, the resulting solution was added dropwise to six different

antisolvents to precipitate nylon particles (S1 Fig in S1 File). The precipitated particle size was apparently different depending on the antisolvent. In the case of water, large clumps were formed. Other solvents, such as acetone, EtOAc, and methanol, produced rough particles with visible sizes. The best results were obtained when ethanol and EG were used, and the invisible fine particles were dispersed. The D50 of nylon-6 was approximately 9.8 μm (Fig 1). These microparticles were stably dispersed without aggregation, even though the solvent was displaced to buffer A. NylC-GYAQ-catalyzed enzymatic hydrolysis was performed on precipitated particles. The results of TLC analysis for detecting the released nylon-6 oligomer are shown in Supplemental S2 Fig in S1 File. For the nylon pellet, which was the untreated material, and the dispersed sample in water were detected no visible spots after 24 hr of enzymatic reaction. For the other five solvents, however, apparent spots corresponding to the Ahx monomer and dimer were detected and confirmed the progress of nylon hydrolysis by NylC-GYAQ.

The monomeric rate was calculated from the amino group concentration of released nylon oligomers in the reaction mixture (Fig 2). Before the enzymatic reaction, the percentage of monomer in the reaction mixture was estimated only around zero. After 72hr enzymatic hydrolysis, the monomeric rates increased depending on the antisolvent used. The values were as follows; 0.4 ± 0.2% for water-resuspended nylon-6, 2.1 ± 0.1% for EtOAc, 2.8 ± 0.3% for acetone and 5.4 ± 0.3% for MeOH. The best results were the case of EG or EtOH-resuspended nylon-6, the monomeric rate showed 7.7 ± 0.4%. Since the value toward pellet was only 0.3% even after enzyme treatment, it was confirmed that homogeneous dispersion brings to increase the sensitivity to NylC-GYAQ catalyzed hydrolysis. In addition, when the NylB-DNY enzyme was coexisted with NylC-GYAQ in the dispersed EG sample, the monomeric rate showed 11 ± 0.2%. This enhancement is attributed to the hydrolysis of dimers to monomers by NylB-DNY, which cannot be hydrolyzed by NylC-GYAQ.

This method is also effective for nylon-6,6. By utilizing EG as suspending solvent, homogeneous dispersion of nylon-6,6 was investigated. Finely dispersed nylon-6,6 was obtained, and its D50 value was 46.8 μm (Fig 1). These microparticles were also stably dispersed without aggregation in buffer A. After 72 hr treatment by Nyl series enzymes (NylC-GYAQ and NylB-DNY), monomeric rate was reached to 14 ± 0.2%. The value toward pellet was limited in

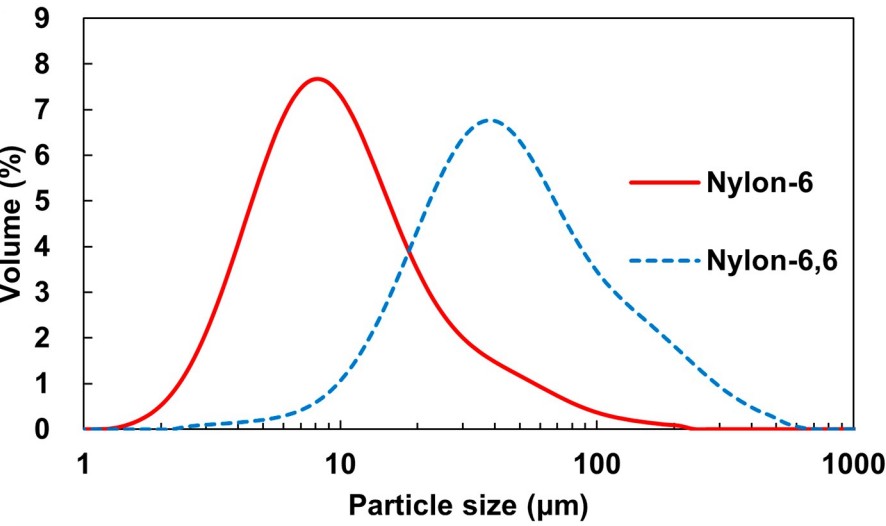

**Fig 1. The particle size of homogeneous dispersed nylons.** -Nylon-6, ···Nylon-6,6.

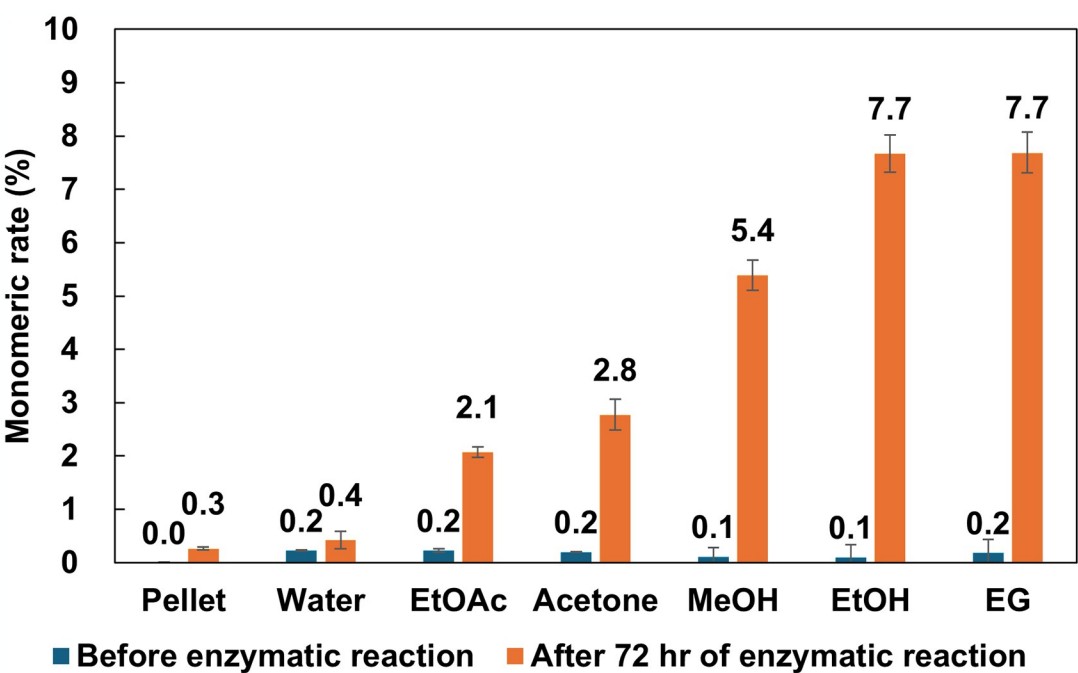

**Fig 2. Monomeric rate before and after enzyme reaction of homogeneous dispersed nylon-6.** The experiments were conducted in triplicate (N = 3).

1%. Through the homogeneous dispersion techniques, we achieved over 10-fold greater enzymatic hydrolysis sensitivity of the nylon-6 and nylon-6,6.

## Formic acid caused soluble oligomerization of polymeric nylons

As an alternative approach for ensuring the water solubility of nylons, we explored a method for lowering their molecular weight. For subdividing polymeric nylons to smaller fragments, soluble oligomerization using formic acid was adopted. The nylon-6 concentration was set at 150 g/L. After 96 hr treatment at 75°C, 100°C, or 120°C, formic acid was removed by lyophilization, and the resulting samples were resuspended in buffer A (S3 Fig in S1 File). Although modest sedimentation occurred in the 75°C- and 100°C-treated samples, no sedimentation was observed in the 120°C sample, and the nylon dispersed finely in buffer A. The hydrolytic reactions by Nyl series enzymes were subsequently performed on these nylon samples. With respect to the TLC analysis of 75°C-treated samples, no spots were observed before the enzymatic reaction, and only weak spot of Ahx monomer appeared after enzyme treatment (S4 Fig in S1 File). At 100°C, however, some weak spots corresponding to the Ahx monomer, dimer, trimer, or other oligomers were observed, and apparent Ahx monomer spots could be detected after the enzymatic reaction. At 120°C, obvious oligomer spots were detected even before the enzymatic reaction, and they were converted into monomers by the Nyl series treatment. The monomeric rates, calculated based on the TNBS assay, are summarized in Fig 3. The calculated percentage of monomers after soluble oligomerization (before enzymatic reaction) were 7 ± 0.1% at 75°C, 12 ± 0.2% at 100°C, 20 ± 0.8% at 120°C, respectively. After enzymatic reaction, monomeric rates were 9 ± 0.2% at 75°C, 41 ± 0.5% at 100°C, and 68 ± 1.4% at 120°C, respectively. The values were improved at 100°C or 120°C-treated samples, although no significant change was observed at 75°C-treated sample. In addition, prolonging the treatment time

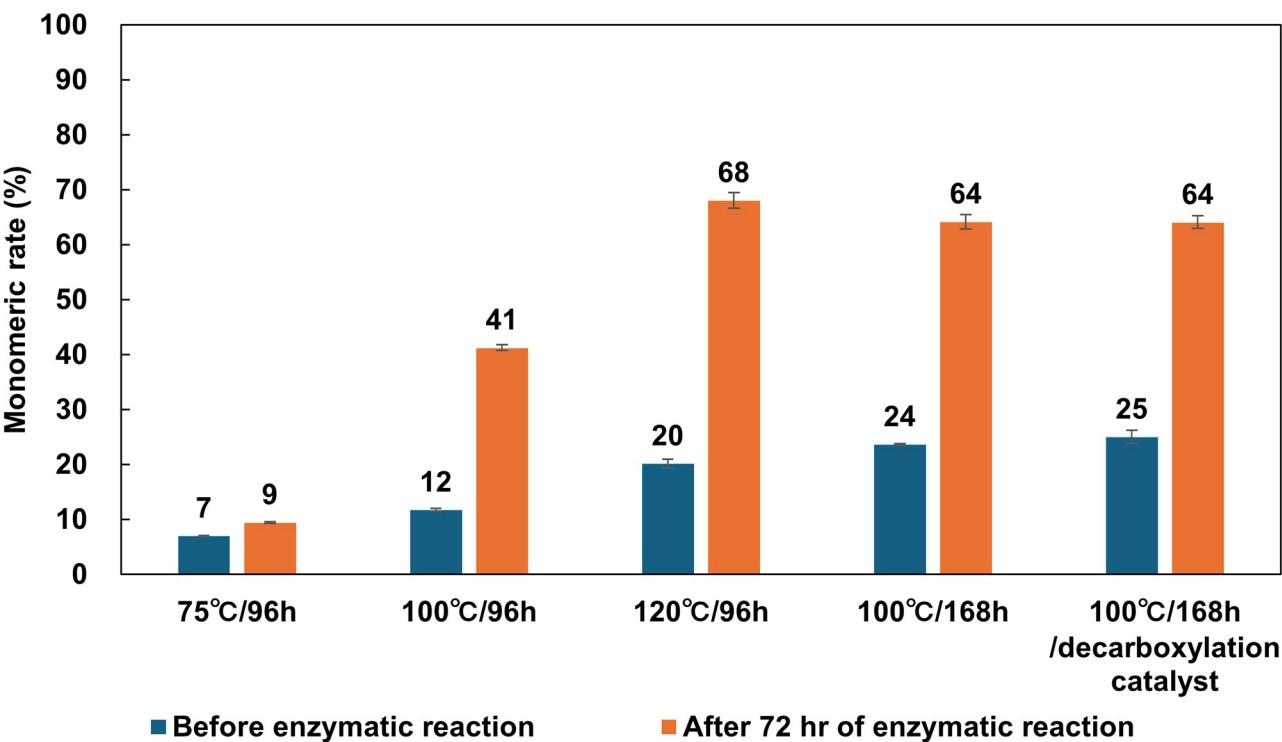

**Fig 3. Monomeric rate of formic acid caused soluble oligomerization of nylon-6 before and after Nyl series catalyzed enzymatic hydrolysis.** The experiments were conducted in triplicate (N = 3).

to 168 hr toward 100˚C-treated sample improved the percentage of monomers to 24 ± 0.2% after soluble oligomerization and 64 ± 1.3% after enzymatic hydrolysis. By doubling the treating time of soluble oligomerization, the monomeric rate at 100˚C sample was equivalent to that of the 120˚C treated sample.

In case of nylon-6,6, the soluble oligomerization underwent at 100˚C for 168 hr. TLC analysis recognized several spots after the acid treatment (S5 Fig in S1 File). These were identified as a monomer unit of nylon-6,6, adipoyl-hexamethylendiamine (6,6-MU), and 1.5-mer of 6,6-MU with diamine terminal, adipoyl-bis[hexamethylenediamine] (6,6N-1.5mer). By treating Nyl series enzymes, these spots disappeared, and were converted into HMD and AD. The monomeric rate was 15 ± 0.6% after acid treatment and reached to 70 ± 5.4% after enzymatic degradation.

To establish the cost-effective removal method of formic acid, a formic acid decarboxylation catalyst was used.[18] The formic acid disappeared within a few hours only by the incubation with a decarboxylation catalyst, resulted in a nearly neutral aqueous solution. By subjecting the enzymatic hydrolysis using a Nyl series, it was revealed that monomeric rate achieved to be an equivalent the case of lyophilization process (Fig 3).

## Combination of homogeneous dispersion and formic acid caused soluble oligomerization

Motivated by the good results of homogeneous dispersion and soluble oligomerization toward polymeric nylons, a combination of these two methods was investigated for achieving a higher monomeric rate. The nylon pellets were subjected to homogeneous dispersion in EG as a

**Table 2. Nyl series catalyzed enzymatic monomeric of commercially available nylon after various chemical pretreatment.**

| | Chemical pretreatment | Nylon-6 | | | | Nylon-6,6 | | | |
|---|---|---|---|---|---|---|---|---|---|
| | | Monomeric rate [%] | Mw | Mn | Mw/Mn | Monomeric rate [%] | Mw | Mn | Mw/Mn |
| 1 | No treatment (pellet) | 1±0 | 102,050 | 18,180 | 5.6 | 2±0 | 102,680 | 14,250 | 7.2 |
| 2 | Homogeneous dispersion | 11±0.2 | 67,730 | 8,090 | 8.4 | 14±0.2 | 69,050 | 8,050 | 8.6 |
| 3 | soluble oligomerization | 64±1.3 | 3,050 | 1,280 | 2.4 | 70±5.4 | 1,870 | 860 | 2.2 |
| 4 | Combined methods 2 and 3 | 95±1.3 | 590 | , 370 | 1.6 | 95±0.6 | 1,490 | 700 | 2.1 |

The experiments were conducted in triplicate (N = 3).

antisolvent, followed by formic acid caused soluble oligomerization at 100°C for 168 hr. Toward these chemical pretreatment nylons, Nyl series- catalyzed enzymatic hydrolysis was performed (S6 Fig in S1 File). TLC analysis indicated that strong spots corresponding to oligomers, especially the Ahx monomer and dimer, were detected in nylon-6 sample, and the 6,6-MU and 6,6 N-1.5mer were identified in the nylon-6,6 sample. The addition of the Nyl series caused these spots to converge into Ahx monomer spots in nylon-6 and HMD as nylon-6,6, respectively. The calculated percentage of monomers after chemical treatment reached 69 ±2.6% for nylon-6 and 50±0.8% for nylon-6,6. After the enzymatic reaction, the monomeric rate reached 95% for both nylons (Table 2).

The changes in the molecular weights of the chemically treated nylons were calculated via GPC analysis (Table 2 and S7 and S8 Figs in S1 File). After homogeneous dispersion, the molecular weights of both nylon-6 and nylon-6,6 unchanged. However, soluble oligomerization by formic acid decreased the average molecular weight and increased the contents of

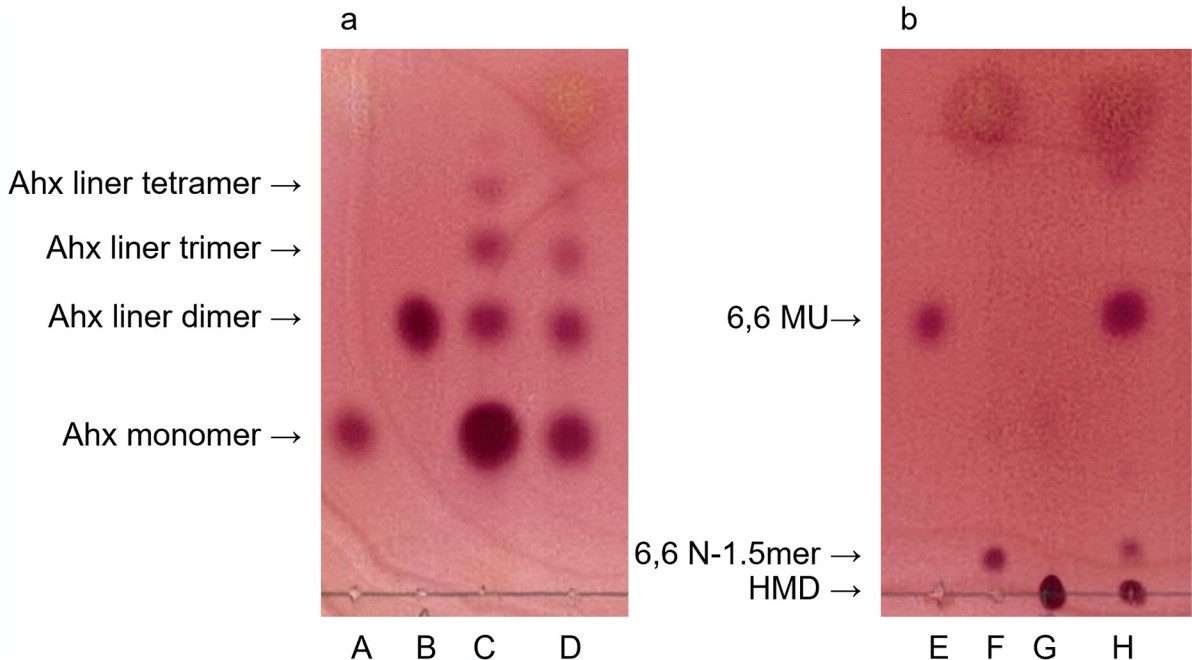

**Fig 4. TLC analysis after 18% hydrochloric acid caused soluble oligomerization (before enzymatic reaction) of nylon-6 (a) and nylon-6,6 (b).** A: Ahx (monomer), B: Ald (Ahx linear dimer), C: Ahx linear oligomer, D: Sample, E: 6,6-MU, F: 6,6N-1.5mer, G: HMD, H: Sample.

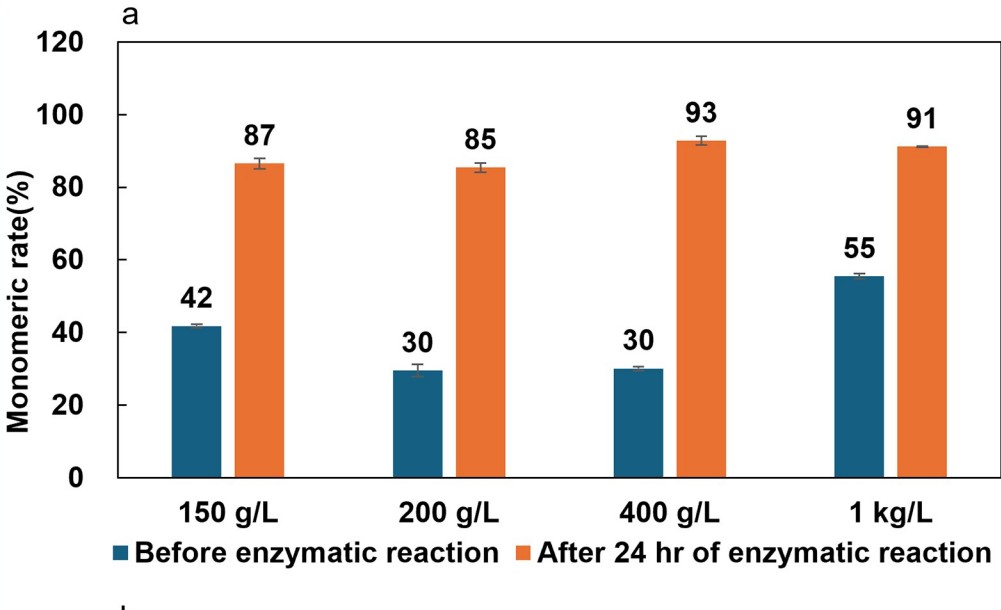

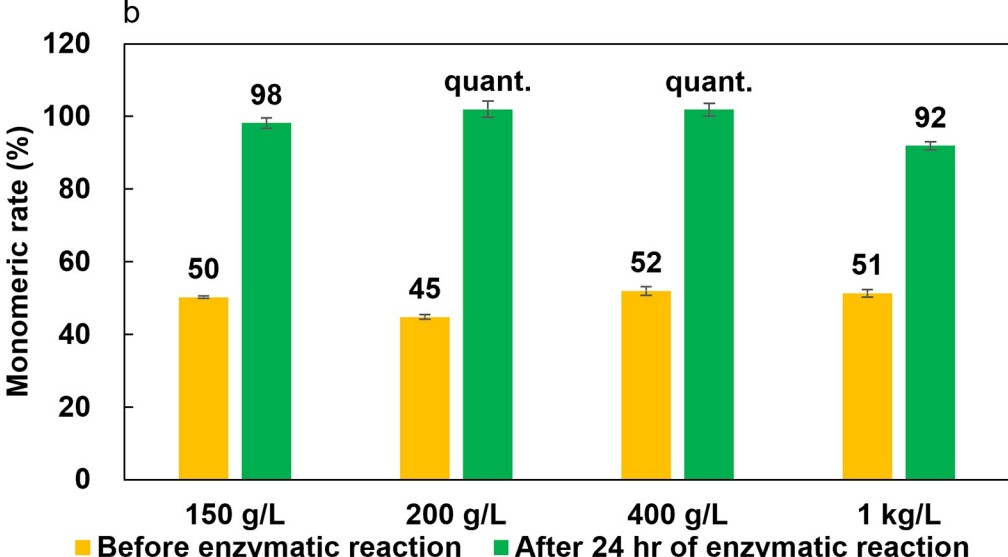

**Fig 5. Monomeric rate of 50% sulfuric acid (a) or 18% hydrochloric acid (b) caused soluble oligomerization of nylon-6 before and after Nyl series catalyzed enzymatic hydrolysis.** The experiments were conducted in triplicate (N = 3).

oligomers. A significant decrease in molecular weight was observed the sample after combination of homogeneous dispersion and soluble oligomerization. Their average molecular weight were 590 for nylon-6 and 1,490 for nylon-6,6, respectively. These molecular weights correspond to Ahx pentamer and the 6,6-MU hexamer. By reducing the average molecular weight, the water solubility of the chemical pretreatment nylon molecules improved, rendering them susceptible to enzymatic degradation and allowing them to achieve a quantitative monomeric rate. The usefulness of the formic acid decarboxylation catalyst has also been confirmed in this process.

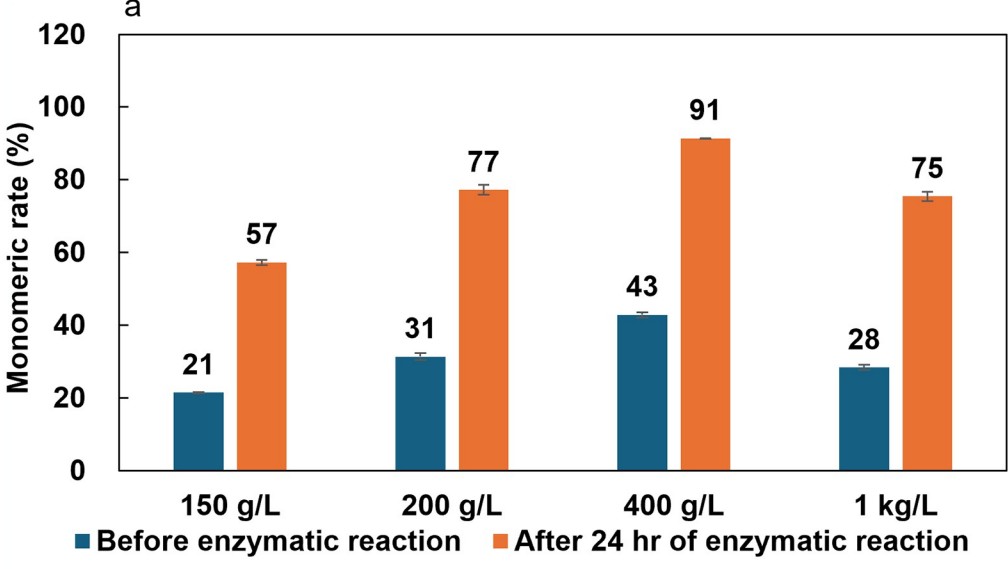

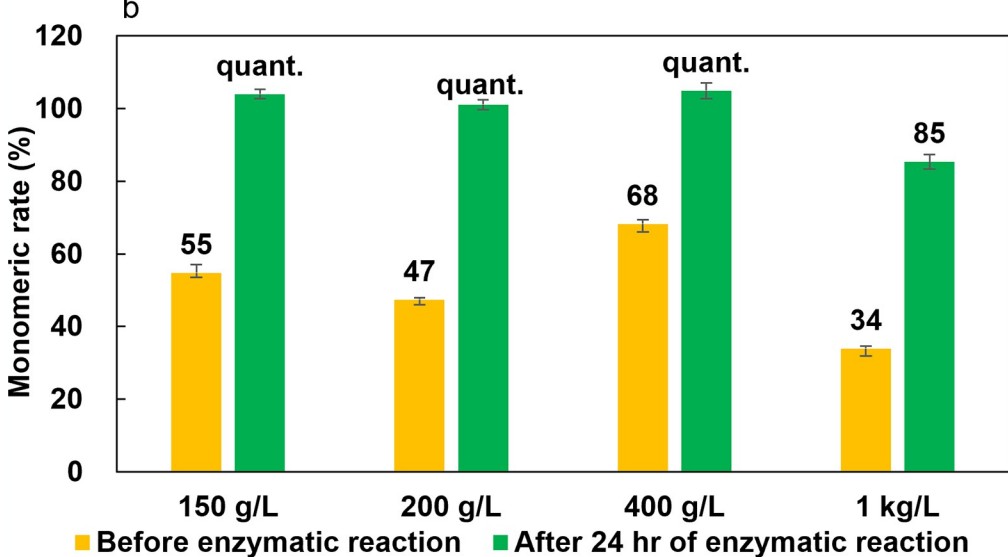

**Fig 6. Monomeric rate of 50% sulfuric acid (a) or 18% hydrochloric acid (b) caused soluble oligomerization of nylon-6,6 before and after Nyl series catalyzed enzymatic hydrolysis.** The experiments were conducted in triplicate (N = 3).

## Other acids caused soluble oligomerization

Shukla and co-workers had attempted to depolymerize nylon-6 at 40 g/L concentration toward hydrochloric acid or sulfuric acid treatment [19]. They used a 30–60% aqueous solution of concentrated hydrochloric acid (effective hydrochloric acid concentration 9–18%) or a 15% aqueous solution of sulfuric acid, and reacted at reflux temperature for 4–8 hr. They indicated that nylon-6 was carbonized under the >20% sulfuric acid concentration. They had also predicted that the solution after soluble oligomerization would consist only of water-soluble compounds such as Ahx monomer and soluble oligomer compounds, because no precipitation

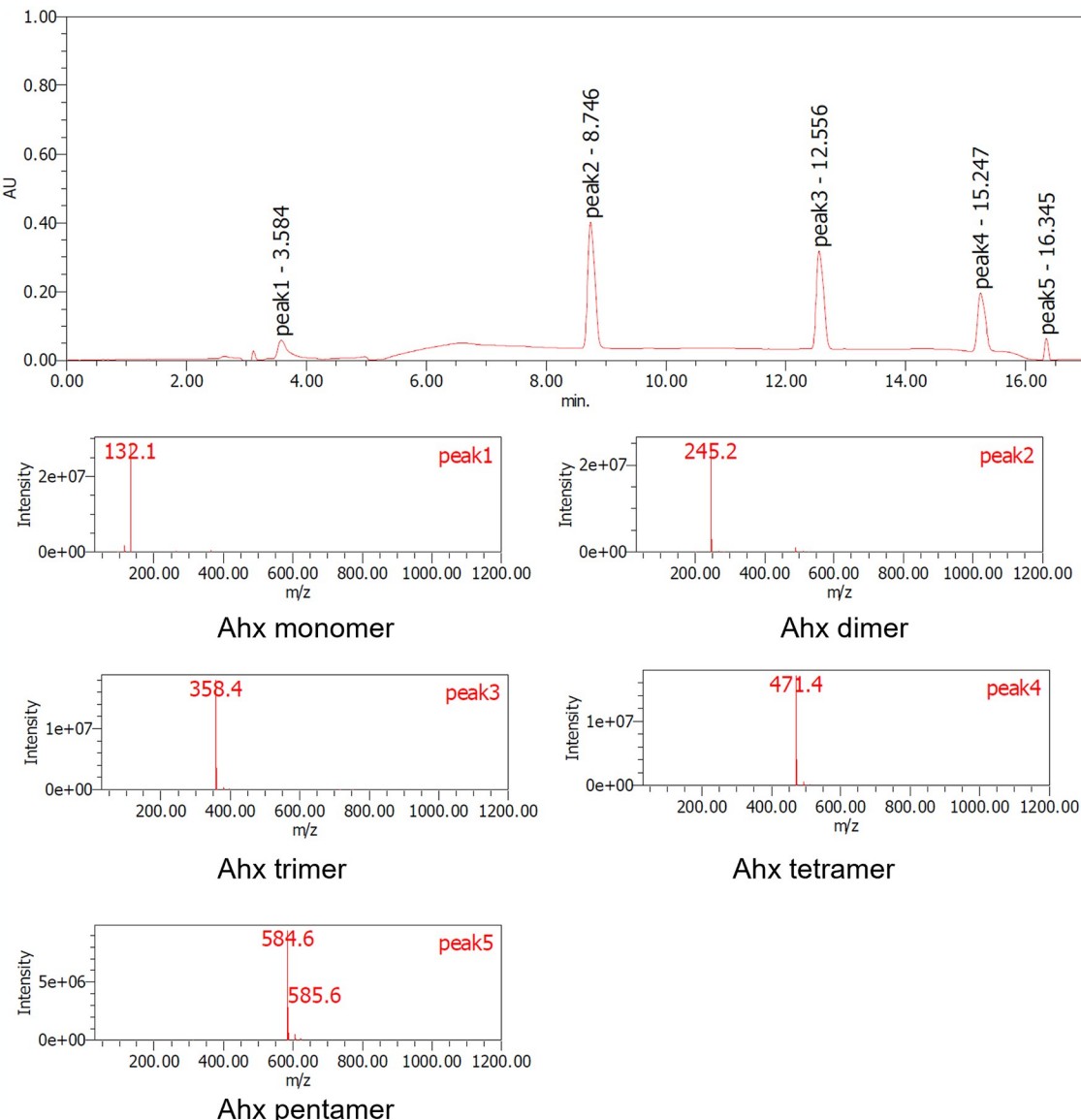

**Fig 7. LC-MS analysis of nylon-6 before Nyl series treatment (soluble oligomerized sample induced by 18% hydrochloric acid treatment).** peak1: Ahx monomer, peak2: Ahx dimer, peak3: Ahx trimer, peak4: Ahx tetramer, peak5: Ahx pentamer.

occurred by the addition of distilled water to the acid treated polymer solution, and the solution was optically transparent.

Firstly, we investigated that nylon-6 pellets were subjected to a 50% aqueous hydrochloric acid (effective concentration is 18%) or a 50% aqueous sulfuric acid at 150 g/L concentration, which concentration was the same with our formic acid treatment described above section, and heated in an oil bath at 120°C for 2 hr. The pellets completely dissolved within an hour, although the reaction mixture did not reflux at this heating. Differently from Shukla's report, nylon-6 did not carbonize even under 50% aqueous sulfuric acid. Interestingly, the pellets completely dissolved within 2 hr even when the concentration of nylon-6 was increased to

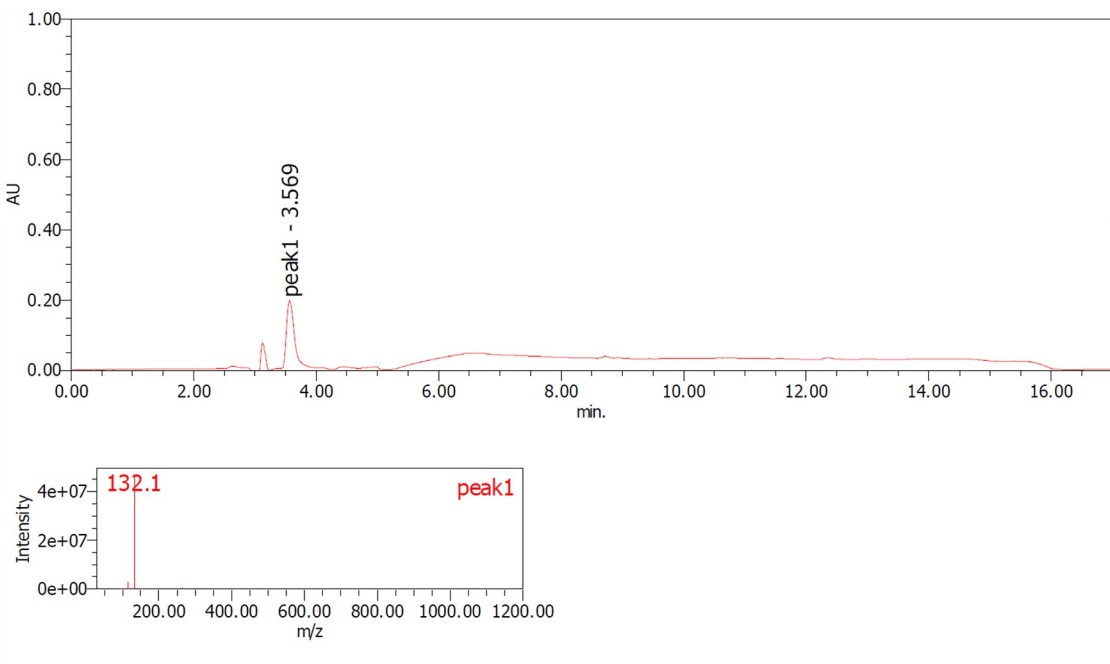

**Fig 8. LC-MS analysis of nylon-6 after 18% hydrochloric acid treatment followed by Nyl series hydrolysis.** peak1: Ahx monomer.

200, 400, and 1,000 g/L. At higher concentrations (> 1,000 g/L), however, some insoluble debris were observed. TLC analysis showed that oligomers remained in the reaction system after chemical processes (Fig 4a). The calculated percentage of monomers after hydrochloric acid and sulfric acid treatment is also higher than that of formic acid caused soluble oligomerization (Fig 5). These oligomers were confirmed to be converted into monomers by Nyl series catalyzed hydrolysis step, and the extremely high monomeric rates (>90%) were achieved.

When nylon-6,6 was treated with 50% sulfuric acid, some insoluble debris had remained even at 150 g/L concentration and monomerization rates remained low (Fig 6a). In the case of 50% hydrochloric acid treated method, however, complete dissolution of pellet was observed under concentrations up to 400 g/L. Dissolved residues were observed at 1,000 g/L concentration. Although several oligomer spots were detected by TLC analysis after soluble oligomerization, HMD was accumulated by Nyl series enzymes treatment and quantitative monomerization achieved (Figs 4b and 6b). Only the case of 1,000 g/L concentration, its value was decreased to 85%.

These results indicate that the complete monomerization can be achieved by adding the enzymatic hydrolysis step using the Nyl series although oligomers remain in only the chemical treatment step, and both nylon-6 and -66 can be efficiently depolymerized even at concentrations as high as 400–1,000 g/L. In addition, it became clear that sufficient chemical degradation proceeds even with pellet feedstock using hydrochloric acid or sulfuric acid in soluble oligomerization treatment step, eliminating the need for an initial homogeneous dispersion process.

The samples, both before and after acid and/or enzyme treatments, were analyzed using TGA, LC-MS, and 1H-NMR techniques. S9 and S10 Figs in S1 File. present the TGA data for

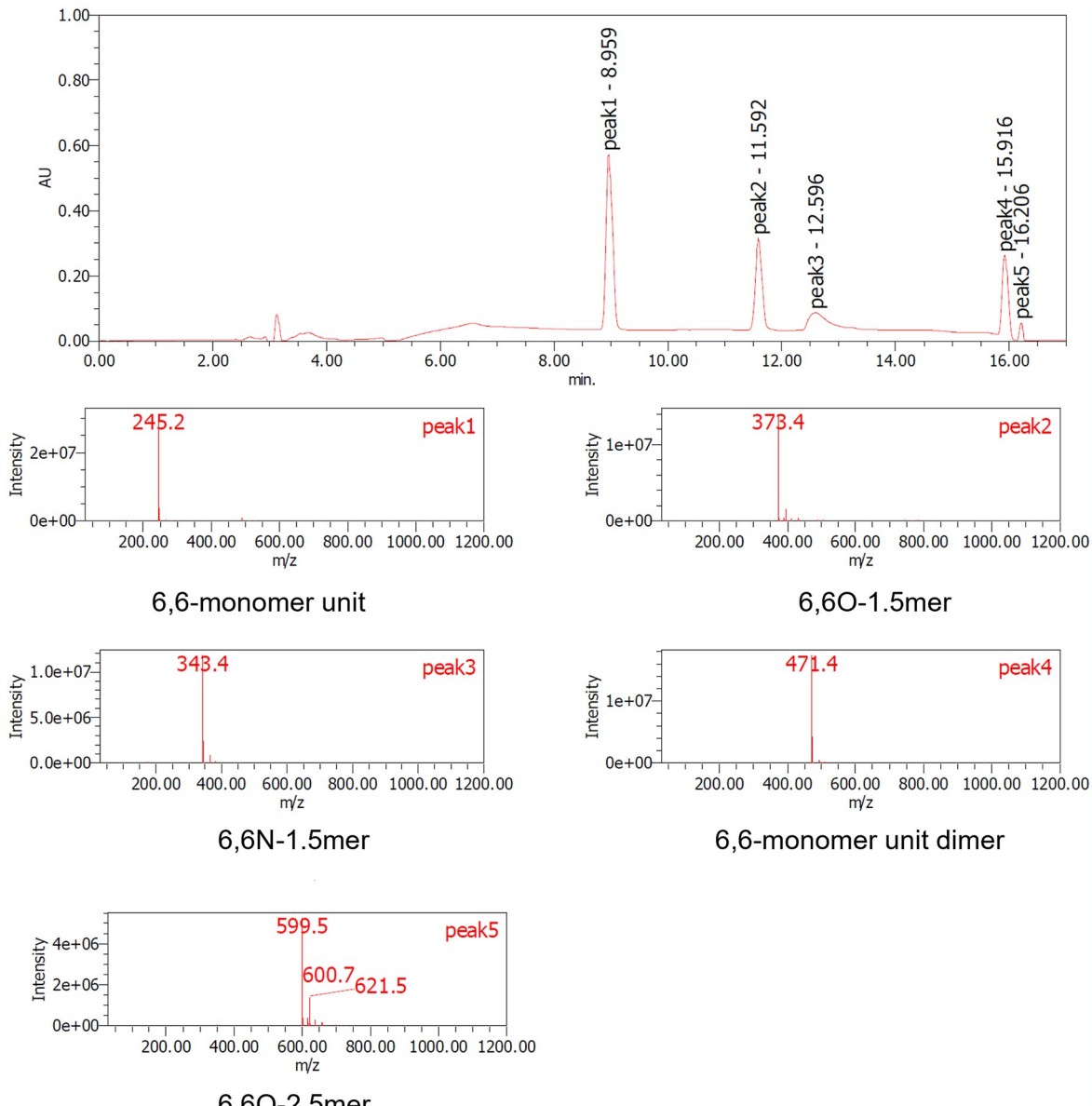

**Fig 9. LC-MS analysis of nylon-6,6 before Nyl series treatment (soluble oligomerized sample induced by 18% hydrochloric acid treatment).** peak1: 6,6-monomer unit, peak2: 6,6O-1.5mer, peak3: 6,6N-1.5mer, peak4: 6,6-monomer unit dimer, peak5: 6,6O-2.5mer.

nylon pellets and the acid caused solubilized samples. The results reveal distinct differences in thermal decomposition behavior between the original nylon pellets and the acid caused solubilized samples. While the nylon pellets decomposed around 400°C, the acid caused solubilized samples exhibited a three-step decomposition, with the first stage beginning at approximately 200°C. This initial degradation aligns with the behavior observed for nylon monomers such as Ahx and AD-HMD salt. The second stage of degradation occurred at around 400°C, consistent with the thermal behavior of high molecular weight nylons. The constant residue observed between 500°C and 800°C was attributed to NaCl, generated during the neutralization process.

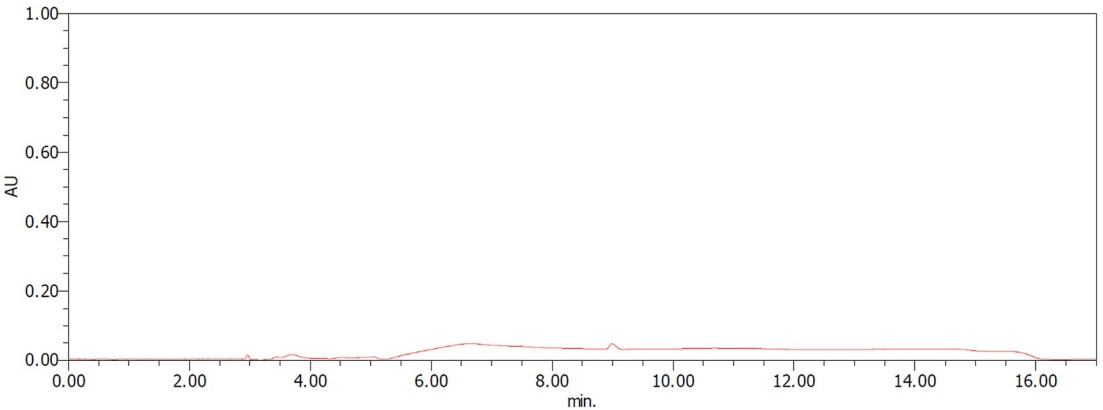

A

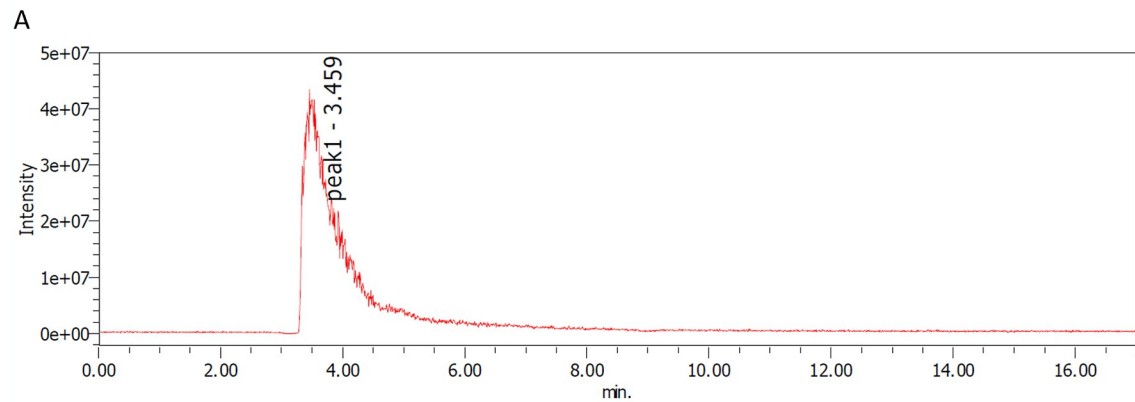

Hexamethylenediamine (117)

B

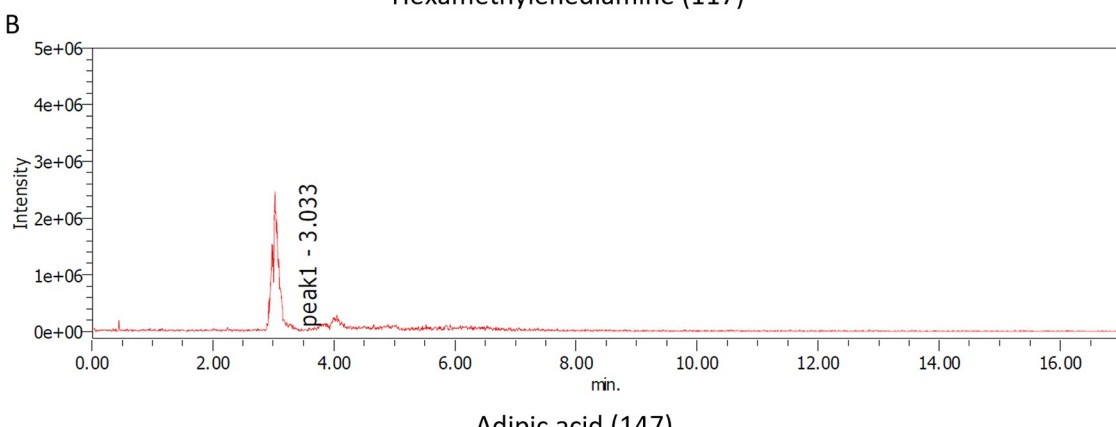

Adipic acid (147)

**Fig 10. LC-MS analysis of nylon-6,6 after 18% hydrochloric acid treatment followed by Nyl series hydrolysis.** A: MS positive scan of 117 (HMD), B: MS positive scan of 147 (AD).

These findings suggest that the acid caused solubilized samples contain significant amounts of monomeric and oligomeric components. Next, the molecular species present in the reaction mixture were tracked using LC-MS analysis (Figs 7–10). After acid caused soluble oligomerization, it was confirmed the presence of monomers and some oligomers in the reaction. The oligomers disappeared by subjecting the Nyl series enzymes and were converting into

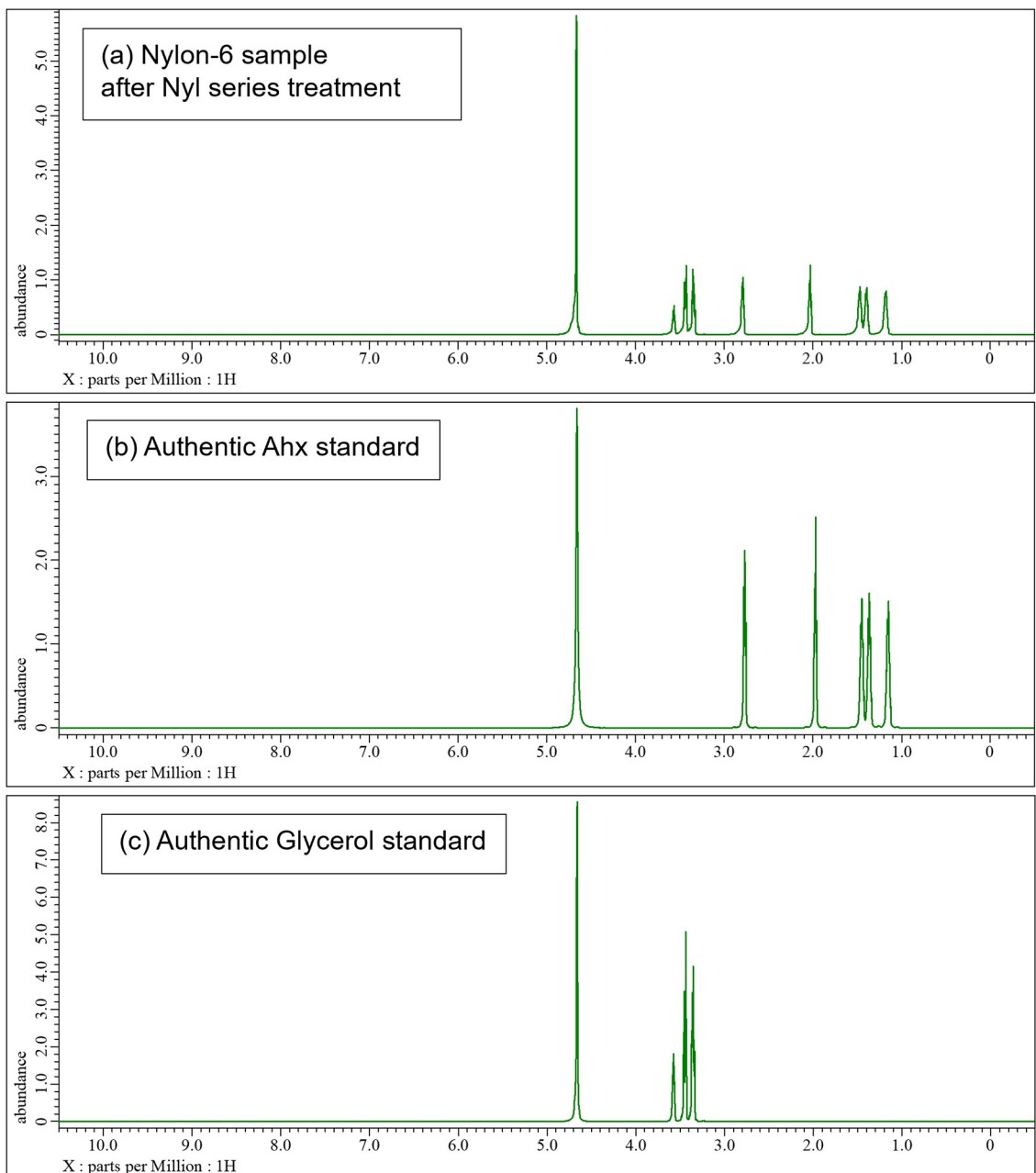

**Fig 11. 1H-NMR analysis of nylon-6 after 18% hydrochloric acid treatment followed by Nyl series hydrolysis.** A: nylon-6 sample after hydrochloric acid treatment followed by Nyl series hydrolysis, B: Authentic Ahx monomer, C: Authentic Glycerol.

monomer units such as Ahx for Nylon-6, or HMD and AD for Nylon-6,6, respectively. Additionally, the 1H-NMR spectra of Nyl series treated samples matched the standards and showed only peaks corresponding to the respective monomer units and glycerol, which was included in BufferA (Figs 11 and 12). These results demonstrate that the established chemo-enzymatic monomerization process can effectively convert high-molecular-weight nylon into monomers without any by-products.

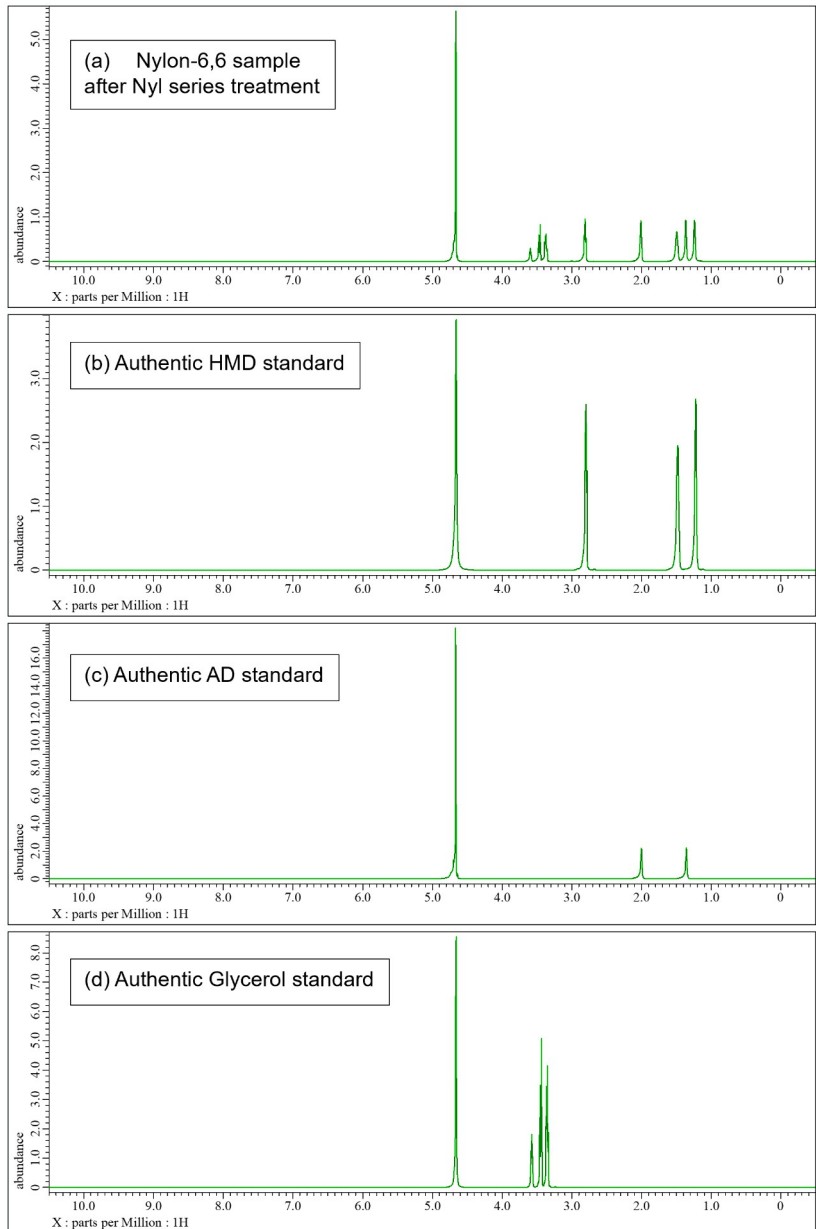

**Fig 12. 1H-NMR analysis of nylon-6,6 after 18% hydrochloric acid treatment followed by Nyl series hydrolysis.** A: nylon-6,6 sample after hydrochloric acid treatment followed by Nyl series hydrolysis, B: Authentic HMD, C: Authentic AD, D: Authentic Glycerol.

Finally, to assess the environmental impact, the Environmental (E) Factor (kg waste / kg product) was calculated [25]. The values were 1.1 for nylon-6 and 1.3 for nylon-6,6, respectively. The ideal E Factor is zero, with higher values indicating more waste and, consequently, a greater negative environmental impact. While a more detailed investigation into the applicability of this value will be necessary following scale-up studies, the calculated value is thought to be acceptable for a chemical recycling of nylon on a laboratory scale.

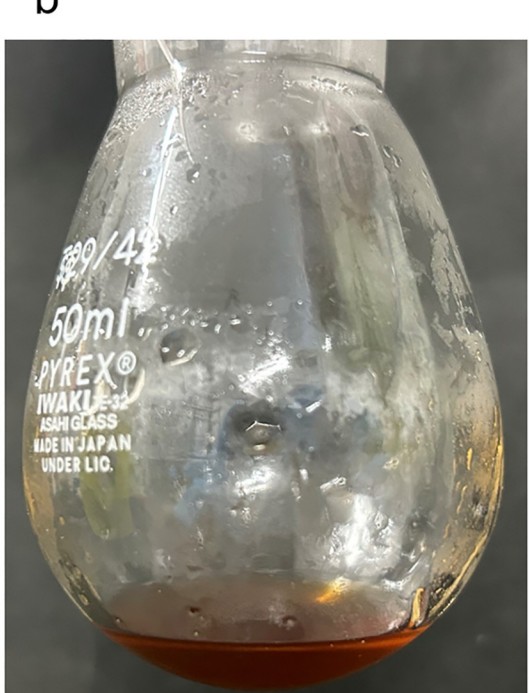

**Fig 13. Photographs of inside the flask before (a) and after (b) 18% hydrochloric acid caused soluble oligomerization toward used fishing net.**

### Verification of applicability toward used fishing net

The applicability of our established nylon monomerization method was tested on fishing nets that had been used as bottom trawl fishing for one year (January 2023 to January 2024) at Shibushi bay in Kagoshima, Japan. The soluble oligomerization was carried out in a 50% aqueous hydrochloric acid (effective concentration is 18%) at 120˚C for 2 hr (Fig 13). The nets were completely dissolved within 30 minutes. After enzymatic hydrolysis with Nyl series enzymes, it was clear that this net was made of nylon-6 and the monomerization rate was 80% (S11 Fig in S1 File). Co8nsidering that it was compounded with other materials for commercialization and non-nylon adhering materials on the surface due to environmental exposure, it was confirmed that our method can be also applicable to market products without issues.

### Conclusion

In this article, we achieved the quantitative monomerization of polymeric nylons utilizing Nyl series enzymes following chemical pretreatment. Investigated chemical pretreatment methods were homogeneous dispersion and soluble oligomerization induced by acid treatment. Homogeneous dispersion increases water solubility of nylon while maintaining its high molecular weight. In contrast, soluble oligomerization enhances water solubility by reducing the molecular weight, leading to oligomerization. In particular, soluble oligomerization using some acids proved effective in increasing the enzyme sensitivity of the Nyl series and achieving the complete monomerization of nylon-6 and nylon-6,6. By fully converting

the oligomers produced through chemical treatment into monomers via enzymatic hydrolysis, the purities of the regenerated monomers were improved and expected to enhance the efficiency of the repolymerization process. This approach is not feasible with chemical methods alone. While there are still many factors to consider in the future, such as techno-economic analysis (TEA), life cycle assessment (LCA), and scale-up studies to confirm the economic feasibility of the method, this study has demonstrated the effectiveness of our approach on market products such as used fishing nets. We believe this underscores the significant potential of our method for achieving full-scale chemical recycling of nylon products in the near future.

## Supporting information

**S1 File. Information on enzyme preparation and S1 to S11 Figs with their respective captions.**
(PDF)

## Author Contributions

**Conceptualization:** Yuki Shiraishi, Dai-ichiro Kato.

**Data curation:** Yuki Shiraishi, Dai-ichiro Kato.

**Formal analysis:** Yuki Shiraishi, Dai-ichiro Kato.

**Funding acquisition:** Dai-ichiro Kato, Seiji Negoro.

**Investigation:** Yuki Shiraishi, Kaito Miyazaki, Maina Yonemura, Yoko Furuno, Risa Yokoyama, Yukiko Yokogawa, Sho Nonaka, Yoshiro Kaneko.

**Methodology:** Yuki Shiraishi, Dai-ichiro Kato.

**Project administration:** Dai-ichiro Kato.

**Resources:** Keigo Ebata, Yuichiro Himeda.

**Supervision:** Dai-ichiro Kato, Seiji Negoro.

**Validation:** Yuki Shiraishi.

**Visualization:** Dai-ichiro Kato.

**Writing – original draft:** Yuki Shiraishi, Dai-ichiro Kato.

**Writing – review & editing:** Yuki Shiraishi, Dai-ichiro Kato, Yoshiro Kaneko, Keigo Ebata, Yuichiro Himeda, Seiji Negoro.

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
