## [Decision Letter · Decision Letter 0]

23 Aug 2024

PONE-D-24-29644Quantitative nylon monomerization by the combination of chemical pretreatment and enzymatic hydrolysis using nylon hydrolasesPLOS ONE

Dear Dr. Kato,

Thank you for submitting your manuscript to PLOS ONE. After careful consideration, we feel that it has merit but does not fully meet PLOS ONE’s publication criteria as it currently stands. Therefore, we invite you to submit a revised version of the manuscript that addresses the points raised during the review process.

We look forward to receiving your revised manuscript.

Kind regards,

Leonidas Matsakas

Academic Editor

PLOS ONE

**Journal Requirements:**

This work was partly supported by a Moonshot Research and Development Program (JPNP18016) commissioned by the New Energy and Industrial Technology Development Organization and Supporting program for interdisciplinary projects in Kagoshima University (DK), and a grant-in-aid for scientific research [Japan Society for Promotion of Science, No. 19K05171 (SN)].

Reviewers' comments:

Reviewer's Responses to Questions

**Comments to the Author**

1. Is the manuscript technically sound, and do the data support the conclusions?

Reviewer #1: Partly

Reviewer #2: Partly

Reviewer #3: Yes

Reviewer #4: Yes

2. Has the statistical analysis been performed appropriately and rigorously? 

Reviewer #1: N/A

Reviewer #2: No

Reviewer #3: N/A

Reviewer #4: No

3. Have the authors made all data underlying the findings in their manuscript fully available?

Reviewer #1: Yes

Reviewer #2: Yes

Reviewer #3: Yes

Reviewer #4: Yes

4. Is the manuscript presented in an intelligible fashion and written in standard English?

Reviewer #1: Yes

Reviewer #2: No

Reviewer #3: Yes

Reviewer #4: Yes

5. Review Comments to the Author

**Reviewer #1:** The manuscript provides new scientific insights into the degradation of polyamides (nylon) with enzymes (nylonases). To this date, the recycling and degradation of these polymer remains a major problem. Although there are a few somewhat effective chemical methods to degrade these polymers, as the authors mention correctly: “the degradation efficiency of enzymes for bulk polymeric nylon is limited to only a few percent at best.” In that aspect, the results provided in this manuscript are a giant leap forward, since they were able to completely depolymerize nylon 6 and nylon 66. However, the following comments/questions should be addressed before I would recommend this article for publication:

1. In the introduction, the authors state “Nylon-6 and nylon-6,6 are composed of simple monomer units such as 6-aminohexanoate (Ahx) for nylon-6 and hexamethylenediamine and adipic acid for nylon-6,6,respectively.” Although I understand that the authors want to introduce the “Ahx” abbreviation, this sentence gives the impression that nylon-6 is made from Ahx while in reality it is caprolactam. I would suggest to introduce a little nuance.

2. In the sentence “NylB is an enzyme that hydrolyzes Ahx oligomers in an exo-type mode, and NylC degrades Ahx cyclic and -linear oligomers with a degree of polymerization greater than three in an endo-type mode” the authors should clarify what they mean with “exo” and “endo”-mode.

3. Nylon-6 and Nylon-6,6 are known to dissolve in formic acid. A) The authors should provide a blank reaction with nylon dissolved in formic acid at room temperature and treat it as if it was hydrolysed at 75-120°C (section 3.1). B) In section 2, Homogeneous dispersion of nylon: is there a reason why polyamides were dissolved in TFE and not formic acid? Formic acid is significantly more environmentally friendly. The authors should at least try to obtain homogeneous dispersions of nylon based on dissolving it in formic acid and adding it to antisolvent. Are they as easily broken down as in the TFE case?

4. In section 3.2, the authors refer to the homogeneous Ir-catalyst as a formic acid decarboxylation catalyst. In my opinion it would be significantly more clear for the reader to refer to it as a dehydrogenation catalyst or formic acid decomposition catalyst, and clearly state that it breaks formic acid down in CO2 and H2.

5. The authors use the term “poor solvent” (e.g. After 72hr enzymatic hydrolysis, the monomeric rates increased depending on the poor solvent used.) I suggest to replace this term with the more correct “antisolvent”.

6. In figure 2, it is revealed that Nylon66 did degrade more extensively in comparison with nylon 6 in their initial runs even though the particle size was bigger. Do the authors have an explanation why?

7. How resistant are the enzymes to formic acid? Are they inhibited by it? Ideally one or two reference reaction are performed with deliberate addition of formic acid. Additionally, polyamide waste is rarely “pure” polyamide. It can contain plasticizers, other plastics, pigments or fillers (e.g. carbon black). Additional experiments to evaluate the effect of these compounds would greatly improve the quality and impact of their research.

8. Correction of grammatical error: “After enzymatic hydrolysis with Nyl series enzymes, it was clear that this net was made by nylon-6 and the monomerization rate was 80%” -> made of nylon-6

9. Although it is unlikely that the authors made other products than the polyamide monomers, what this paper still lacks is some actual molecular proof that they indeed made some Ahx and hexamethylene diamine (the colorimetric method + TLC only proves that something resembling these compounds reacted with the dye). The authors should make an effort to record a mass spectrum or provide an 1H-NMR of the molecules they made. Have the authors any idea how to isolate their products? Effective product isolation is an extremely important factor, especially since the authors believe that “this method holds great potential for realizing full-scale chemical recycling of nylon products in the market.”

**Reviewer #2: **Shiraishi et al describe methods to convert nylon polymers into their monomeric component parts using a tandem chemical and enzymatic process. The manuscript focuses on two methodologies (1) converting nylons into microparticles using the well-understood solvent/non-solvent method, (2) chemically depolymerizing nylons using acids, with the resulting products subjected to enzymatic hydrolysis.

Whilst the recycling of nylons is a highly interesting topic, and the use of enzymes to recycle polymers has gained a lot of interest, the manuscript feels a little misleading in the way the methods used are being depicted. Firstly, the acid treatment is not a pre-treatment or limited, it is converting high molecular weight polymers to 5-mers, this is a significant amount of depolymerization! Secondly, the analytical methods chosen are not sufficient to support the claims made in the manuscript, as TLC based analysis cannot be used to identify “unknown” compounds alone, even if you suspect you know what those compounds are. Whilst the method presents an interesting route to the depolymerization of nylons, the use of enzymes does not seem warranted when it could be that optimizing the acid depolymerization step would be sufficient. The method is suggested to be a cost effective, environmentally friendly solution, however these things are not explicitly calculated or tested, so this cannot be claimed.

Major comments:

- Throughout the manuscript and in the title, the chemical step is described as a “chemical pre-treatment”. This is simply not true; the described method is an acid catalyzed hydrolysis reaction which significantly deconstructs the polymer and is doing the bulk of the depolymerization. To call this a “pre-treatment” step is misleading and incorrect as the resulting mix of small oligomers can no longer be called a nylon polymer. In the PETase field grinding is considered a “pre-treatment” as it does not inherently change the chemical composition of the polymer or the polymer chain lengths, as this method does change both these things this sort of language needs to be revised. This should be updated throughout to describe what is actually being carried out i.e. extensive acid catalyzed depolymerization of the nylon followed by an enzymatic step. In addition, I really dislike the use of “limited hydrolysis” when describing the acid step. The depolymerization by the acid is actually very extensive.

- There is a misconception throughout that the enzymatic depolymerization of nylon is hampered by the effects of substrate crystallinity. This is incorrect. The depolymerization is affected by the strong intermolecular hydrogen bonding pattern between polymer chains. The acid treatment is not affecting the crystalline structure of the nylon at all, it is chemically depolymerizing it, turning the long polymer chains into shorter monomers which then do not have the same hydrogen bonding pattern. These short oligomers have no characteristic of “crystallinity” as they are no longer nylon polymer. This needs to be revised throughout the manuscript.

- Again, for the crystallinity, as no DSC has been carried out on the polymer there is no evidence that the polymer used is highly crystalline. Crystallinity of nylons is also significantly affected by incubation in aqueous buffers, what is your evidence that the nylon in your reactions is significantly crystalline?

- There are numerous grammatical errors throughout the introduction, and phrases which do not make sense like “low simple monomer units”. The introduction needs to be carefully revised to remove these errors and inconsistencies.

- In the introduction, I don’t think it is fair to say that acid hydrolysis increases the water solubility of nylon, it is no longer really a nylon polymer by the time it is significantly hydrolyzed, what you are doing is turning it into more soluble oligomers of nylon.

- For publication in PLOS ONE, the use of TLC as one of the major analysis methods is insufficient. TLC does not tell you what the products are and is not informative enough for the described study. There are far superior analysis methods which accurately detect and quantify nylon oligomers such as mass spectrometry, LC-MS, HPLC and LC-MS/MS. Phrases such as the monomer concentration was “estimated” to be zero are not appropriate when this could be rigorously analyzed.

- The TNBS method is insufficient as an analysis method for this study. It is not shown that the nylon oligomers of different lengths react with TNBS in the same manner, meaning values obtained by this method may be incorrect. The monomeric rate cannot be calculated from the TNBS method as it will also react with nylon oligomers of various sizes, especially in the case of NylC where the major product of the reaction is the dimer, not the monomer. This calculation is misleading and most likely incorrect, as all lengths of oligomer are being treated as “monomer”.

- The use of nylon nano particle suspensions for enzymatic hydrolysis is already described (see https://onlinelibrary.wiley.com/doi/10.1002/anie.202404492), where water was used successfully as the non-solvent. Please could you comment on this discrepancy? This paper should be cited.

- The section on homogenous dispersion is very confusing. From line 149 onwards it is unclear what has been done. Were reactions carried out in solvent? (line 160).

- Using the described analysis methods, the increase in release of monomers cannot be commented on (line 171) as described above.

- Line 177, what is the sedimentation? There is limited characterization of the acid treated polymer, so it is unclear what is in this reaction mixture, what size are the oligomers, what is the ratio of the different oligomer sizes?

- You cannot identify molecules by TLC alone (line 192).

- The chemical catalyst is referred to as “more cost effective” but no cost analysis is done. Iridium is also a precious metal, will this scale well?

- The conclusion claims “quantitative monomerization” but the analysis methods are insufficient to claim this.

- I think it is incorrect to claim this is ready for full-scale chemical recycling of nylon when there has been no TEA/LCA analysis. How would microparticles be created at scale?

- The figure legends are not sufficient to replicate the results, what enzymes were used at what concentration? How were these enzymes expressed and purified?

- The images of the TLCs should be labelled with what the spots are estimated to be.

_ I could no find mention of how many replicates were conducted nor what the error bars on the graph represented.

In conclusion, although this manuscript describes a potentially interesting approach, it requires significant revisions to be of sufficient caliber for publication in PLOS ONE. The analytical approach is quite poor and would need to be significantly revised and updated to support the claims being made, with proper analytics as described above. Furthermore, the misleading claims and language surrounding the acid depolymerization step needs to be removed, and the title updated to reflect this. There needs to be a significant effort to make the language and explanation of what was carried out clearer to avoid confusion. In addition, the figure legends and methods need to be updated so that all experimental details are included and could be carried out by others.

**Reviewer #3:** Quantitative nylon monomerization by the combination of chemical pretreatment and enzymatic hydrolysis using nylon hydrolases

The study by Yuki Shiraishi and colleagues investigates the chemo-enzymatic depolymerization of polyamides 6 and 6,6. The authors compare different chemical pretreatments and the effects on physical properties of the polymers (Mw, Mn, particle size, etc.) and the subsequent enzymatic hydrolysis efficiency. Firstly, the authors investigate the effect of dispersing the nylon (in 6 different solvents) and of limited hydrolysis (using different acids, temperatures and reaction times), including the combination of the two chemical pretreatments, which allowed to increase the enzymatic degradation to 95% for both nylon types. Using stronger acids such as hydrochloric acids allowed to skip the dispersion step and shortened the pretreatment reaction time to only 2 h, compared to the 168h used with formic acid. Finally, the authors tested their process on used fishing nets, reaching 80% monomerization.

Currently, there is a very limited amount of studies dealing with the recycling of real EoL plastic waste, which makes this study of particular interest.

The manuscript is well written and easy to follow. The experimental set up is sound and the results are clearly presented.

The manuscript could be further improved by providing more advanced characterization of the depolymerization products (and the study of the mode of action of the enzyme on these oligomers), and explaining more in detail the enzyme part, so that other researchers could replicate the experiments.

The total lack of explanation regarding the expression and purification of the enzyme is rather striking.

I strongly recommend to either use some references or then explain in the material and methods the protocols used.

Some more detailed comments:

Lines 76-78: how were these enzymes obtained/produced for your experiments?

Are they expressed in E.coli? What protocol for expression and purification? What enzyme yields when operating at increasing nylon titers? Etc.

-Lines 88-89: what about the hexamethylenediamine?

- line 96: please correct “EasIVial” into “EasiVial”

- Line 108: 150 g/L nylon in water?

- Line 212-123: please explain the rational why you used different times of incubations depending on the temperature (Section 3.1) and the type of acid used (2h, 96h, 168h)

- Line 133: please explain better how the enzyme solution was obtained

- line 138: please elaborate better on how you determined the theoretical amino group concentration

- line171: interesting that the larger particle size of nylon 6,6 did not negatively affect the enzymatic depolymerization. This seems to differ from other studies and observations. You might elaborate a bit on this.

- Line 192: how did you identify these monomer units exactly? Did you use specific standards?

- Line 195: as a comparison, the chemical pretreatments with dispersion and limited hydrolysis should be extended for an additional 36-72h (the time of enzymatic hydrolysis incubation) for a better comparison.

This would allow to highlight better the contribution of the chemo-enzymatic compared to the chemical treatment alone.

**Reviewer #4:** Review of manuscript

PONE-D-24-29644

The authors present in their work entitled “Quantitative nylon monomerization by the combination of chemical pretreatment and enzymatic hydrolysis using nylon hydrolases” the combined chemical-biotechnological decomposition of two nylon derivatives, both in form of virgin pellets and one commercial nylon product for recycling purposes.

The work is well structured and written logically, and the description of the methods used for biomass fractionation and compound characterisation are sufficiently detailed such as to allow experts in the field to reproduce eventually. The data seem sound, and the conclusions drawn are justified by the various data presented. A few aspects should nevertheless be addressed during a revision:

• In the introduction, a small table might be interesting detailing the works done in the field with their characteristics, and highlighting the novelty of the current work. Some aspects are cited in the text, but an overview in table form might be more informative as such.

• The number of significant figures used when describing the monomer yields might need to be reconsidered/checked.

• The enzyme loading is relatively high. How easy can the products be isolated, and eventually purified/separated for a renewed use in a hypothetical circular economy setting?

• How many times can the enzymes be technically reused, and how much activity do they lose during the process? How prone are they to be inhibited by impurities like colours, etc, that would be part of a nylon-based clothes, for example?

• The main concern is connected to the real sample treated in this work. The paragraph and the information is rather limited, and some questions remain. For example, why was hydrochloric acid used in this case, and not formic acid? How easy is it to separate the hydrolysis products from the 20% remains? What do the TLCs look like for the real sample?

• Would it be possible to calculate the E-factor for the optimised process, i.e., prehydrolysis plus enzymatic depolymerisation?

6. PLOS authors have the option to publish the peer review history of their article (what does this mean?). If published, this will include your full peer review and any attached files.

Reviewer #1: No

Reviewer #2: No

Reviewer #3: **Yes: **Cristiano Varrone

Reviewer #4: No

---

## [Author Response · Author response to Decision Letter 0]

4 Oct 2024

To the editor and reviewers,

First of all, we would like to express our sincere gratitude to all four reviewers for their insightful and constructive comments. Now we carefully revised our manuscript according to the reviewers’ suggestions. Thanks to reviewers' feedback, we believe we have been able to provide readers with more accurate and detailed information than in the original submission.

To address the revisions, it was necessary to perform additional 1H-NMR analysis. As a result, we would like to propose adding the researcher, Yukiko Yokogawa, who conducted this experiment as a co-author. We have obtained consent from the other co-authors, and we kindly ask for reviewers' consideration on this matter.

We provide our responses to each comment in another pdf file.

---

## [Decision Letter · Decision Letter 1]

8 Dec 2024

PONE-D-24-29644R1Quantitative nylon monomerization by the combination of chemical pretreatment and enzymatic hydrolysis using nylon hydrolasesPLOS ONE

Dear Dr. Kato,

Thank you for submitting your manuscript to PLOS ONE. After careful consideration, we feel that it has merit but does not fully meet PLOS ONE’s publication criteria as it currently stands. Therefore, we invite you to submit a revised version of the manuscript that addresses the points raised during the review process.

We look forward to receiving your revised manuscript.

Kind regards,

Leonidas Matsakas

Academic Editor

PLOS ONE

Journal Requirements:

Reviewers' comments:

Reviewer's Responses to Questions

**Comments to the Author**

1. If the authors have adequately addressed your comments raised in a previous round of review and you feel that this manuscript is now acceptable for publication, you may indicate that here to bypass the “Comments to the Author” section, enter your conflict of interest statement in the “Confidential to Editor” section, and submit your "Accept" recommendation.

Reviewer #2: (No Response)

Reviewer #4: All comments have been addressed

2. Is the manuscript technically sound, and do the data support the conclusions?

Reviewer #2: Yes

Reviewer #4: Yes

3. Has the statistical analysis been performed appropriately and rigorously? 

Reviewer #2: N/A

Reviewer #4: Yes

4. Have the authors made all data underlying the findings in their manuscript fully available?

Reviewer #2: Yes

Reviewer #4: Yes

5. Is the manuscript presented in an intelligible fashion and written in standard English?

Reviewer #2: No

Reviewer #4: Yes

6. Review Comments to the Author

Reviewer #2: The authors have made a good attempt to review the manuscript and have added additional data to support their claims. However, I have a number of suggestions that should be addressed prior to publication:

- In the response to the reviewer’s comments, the authors raise some interesting hypotheses that are not referred to in the manuscript e.g. that amorphous nylon on the surface may affect deconstruction rates, enzyme inhibition by additives and enzyme reuse. These are all valuable points which should be mentioned in the conclusion to highlight where the field is going next. These points are important for understanding the context of the current manuscript.

- The new NMR and LC-MS data is far more compelling evidence of deconstruction than the original TLC images. I strongly suggest the NMR/LC-MS data is moved to a main figure in the manuscript and some of the TLC images can then be moved to the supplementary.

- Even with the calculated E factor, I still think it is too early to suggest this is a scalable, economically feasible and environmentally feasible solution to nylon waste. It would be more honest for the authors to mention in the conclusion that this is an exciting new avenue to explore, but TEA and LCA will need to be used down the line to confirm that it is economically feasible method, and that scaling up will also be important to test in the future.

- I still believe a DSC for the nylons pre- and post- treatment really should be included to help other researchers replicate the research.

- There are still instances of some confusing sentence structures : e.g.“formic acid has a concern to cleave amide bonds”. I strongly suggest the manuscript is carefully checked prior to publication to aid reading clarity.

If the above comments are taken into consideration, I am happy to recommend the manuscript for publication.

Reviewer #4: The authors present the revised version of their work entitled “Quantitative nylon monomerization by the combination of chemical pretreatment and enzymatic hydrolysis using nylon hydrolases”.

The authors revised the manuscript carefully, and responded in a satisfying manner to the reviewers’ comments. The paper seems suitable for publishing.

As a last minor aspect, the authors should check the wording when using the technical term soluble oligomerisation. For example, rather than writing ‘formic acid treated soluble oligomerization’, it might be better to formulate: ‘formic acid caused soluble oligomerisation’, ‘soluble oligomerization upon formic acid treatment’ or ‘soluble oligomerisation induced by formic acid treatment’. Currently one could get the impression that the term has just been added to the sentence.

7. PLOS authors have the option to publish the peer review history of their article (what does this mean?). If published, this will include your full peer review and any attached files.

Reviewer #2: No

Reviewer #4: No

---

## [Author Response · Author response to Decision Letter 1]

8 Jan 2025

To the editor and reviewers,

We would like to extend our sincere gratitude to all reviewers for their insightful and constructive comments. We have carefully revised our manuscript in accordance with the reviewers’ suggestions. Please find the "Response to Reviewers.pdf" file for details.

Thanks to their valuable feedback, we believe the revised manuscript now offers readers more accurate and detailed information compared to the original submission.

Thank you.

DK

---

## [Editor Report · Decision Letter 2]

21 Jan 2025

Quantitative nylon monomerization by the combination of chemical pretreatment and enzymatic hydrolysis using nylon hydrolases

PONE-D-24-29644R2

Dear Dr. Kato,

We’re pleased to inform you that your manuscript has been judged scientifically suitable for publication and will be formally accepted for publication once it meets all outstanding technical requirements.

Kind regards,

Leonidas Matsakas

Academic Editor

PLOS ONE
---

## [Editor Report · Acceptance letter]

23 Jan 2025

PONE-D-24-29644R2 

PLOS ONE

Dear Dr. Kato, 

I'm pleased to inform you that your manuscript has been deemed suitable for publication in PLOS ONE. Congratulations! Your manuscript is now being handed over to our production team.

Kind regards, 

on behalf of

Dr. Leonidas Matsakas 

Academic Editor

PLOS ONE
